# Factors Influencing Saudi Youth Physical Activity Participation: A Qualitative Study Based on the Social Ecological Model

**DOI:** 10.3390/ijerph20105785

**Published:** 2023-05-11

**Authors:** Anwar Al-Nuaim, Ayazullah Safi

**Affiliations:** 1Physical Education Department, Education College, King Faisal University, Al-Ahsa 31982, Saudi Arabia; 2Department of Public Health, Centre for Life and Sport Science (C-LaSS), Birmingham City University, Birmingham B15 3TN, UK

**Keywords:** youth, factors, influencing, physical activity, health, qualitative research, social ecological model

## Abstract

Background: The growing improvement in urbanisation, modes of transportation and the expansion of sedentary behaviour, both at work and home, have resulted in declining rates of physical activity (PA) worldwide. Nearly one-third of the global population aged 15 and over are insufficiently active. The negative effect of physical inactivity has been evidenced and ranked fourth as the lethal cause of death globally. Therefore, the aim of this research was to explore the factors influencing PA participation among youths from different geographical locations in the Kingdom of Saudi Arabia. Methods: Sixteen focus groups (males = 8 and females = 8) were conducted with a total of 120 secondary school students (male = 63 and female = 57) aged between 15 and 19 years. The focus groups were analysed to identify key themes through the process of thematic analysis. Results: Results from the focus groups indicated that a lack of time, safety, parental support, policies, access to sport and PA facilities, and transportation, as well as climate were reported as barriers to PA participation. Discussion and conclusion: The current research contributes to the scarce literature focused on the multidimensional effects on Saudi youth PA behaviour from different geographical locations. This qualitative approach has provided the participants a voice, and the overall study offers valuable evidence as well as invaluable information to policymakers, public health departments, and local authorities for PA intervention based on the environment and the community.

## 1. Introduction

The expansion of urbanisation, modes of transportation, and an increase in the sedentary nature of work and the availability of entertainment gadgets at home have contributed to the decline in physical activity (PA) engagement [1,2]. According to the World Health Organisation (WHO, 2021), more than a quarter of the adult population (1.4 billion adults) and 81% of adolescents aged between 11 and 17 years are insufficiently active. Moreover, girls were reported to be less active than boys, with 85% of boys and 78% of girls failing to meet the WHO recommended guideline of 60 min of moderate to vigorous exercise (MVPA) per day [3]. The negative effect of physical inactivity has been evidenced and ranked fourth as the leading cause of death worldwide [4,5]. Previous research has concluded that engaging in PA is rapidly decreasing in adolescence, especially among females [6]. Furthermore, research reported that in the Kingdom Saudi Arabia (KSA), 92% of the population lead a sedentary lifestyle [2]. The sedentary lifestyle in the KSA is particularly prevalent in females than males [2,7,8]. A sedentary lifestyle is one of the main causes of death and disability across the globe [2]. Therefore, it is important that young people have early exposure to PA as it plays an important role in maintaining a healthy and active lifestyle in adulthood [9,10].

Although the exposure of PA in early age is essential, it must be considered that young people generally do not have a high level of autonomy compared to adults for various reasons, including safety. Consequently, this population has less access to PA facilities, including open parks; thus, the surrounding environment can affect their behaviour to a greater extent than it would to an adult [11]. During the period of adolescence, young people experience increasing independence from parental figures [12,13]. This period in life is also where a young person may obtain a driving licence and start to drive, and consequently, this becomes a transition point for their transport requirements. Gaining access to vehicles can generate a level of apathy, which can negatively affect the individual’s frequencies of walking, cycling, or public transportation use, as a young person progresses into adulthood [14]. Therefore, active travel modes such as walking and cycling and minimising the use of motor vehicles can vastly contribute to health benefits [2,14]. In addition, there is a wider perspective whereby changing and developing lifestyles brought about by new technologies likely increases youth inactivity. National campaigns focused on increasing youth awareness on physical inactivity have reportedly been conducted in developed countries [15,16]. However, while extensive research on this subject has been carried out in most western countries, to the best of our knowledge, no studies have been conducted with respect to the factors influencing youth PA through the social ecological model within Saudi Arabia. There is therefore a need to study and understand the lifestyle factors contributing to the increase in obesity and behaviours that affect the level of PA during the transitional stage in adolescence.

Previous research has shown that a range of studies focusing on PA and health lacked an explanation of the theoretical approaches underpinning their research [17,18]. However, applying a theoretical model can help facilitate an understanding of effective and ineffective methods based on the individual’s intention and reasoning to engage or otherwise in PA, health, or wellbeing programmes [18,19]. To comprehend the outlined reasons behind an individual’s PA engagement, one of the recommended techniques is to recognize and/or apply behavioural change theories such as the health belief model [20], the transtheoretical model [21], the self-determination theory [22], and the social ecological model [23] that are commonly used across disciplines and countries to establish the feasibility and effectiveness of various PA, health, and wellbeing interventions.

Previous research has suggested that the existing literature focused on the influence of social factors has expanded from merely evaluating the psychosocial and demographic factors, and that there is a need for investigating the effect of environmental factors on young people’s PA and health behaviour. Moreover, several theoretical models have assessed the determining factors behind PA or inactivity. For instance, research suggested that most of these are centred on factors based on an individual rather than considering the wider context [24]. For the purpose of this study, the SEM is deemed appropriate to be applied. The SEM of [25] suggests that people’s behaviour is not merely influenced by intrapersonal characteristics, but also by various other social factors. Hence, this model is particularly relevant to this study in the context of both the individual and wider aspects that impact young people’s PA engagement: intrapersonal, interpersonal communities, organisational, and policy/environment. Furthermore, the SEM helps identify opportunities to promote the PA of individuals or groups, determining various factors that may impact their engagement [26].

A large amount of the literature suggests that the lack of PA engagement is linked to SEM [27,28,29,30]. The SEM suggests that people’s behaviour is not merely influenced by intrapersonal characteristics, but also by various social factors. For instance, the SEM has four different levels: (1) intrapersonal, which refers to the characteristics of an individual that influence behavioural change, such as attitude, knowledge, and expectations; (2) interpersonal community, which refers to social networks such as family, friends, co-workers, shared identities, and relationships that may impact behaviour; (3) organisational, which refers to the rules, regulation, and strategies that may promote or endanger health; and (4) policy/environment, which refers to the policies, advocacy, and environmental structures that impact people’s physical activity levels, health, and wellbeing [30,31]. The SEM has been widely used in various disciplines, including nutrition, obesity, PA, and health across developing and developed countries, including China, the UK, Nigeria, and the KSA [32,33,34,35,36,37]

However, individual and contextual factors are parts of an SEM framework in relation to health behaviours, and the effect of these inputs is transactional. For instance, individual factors may include elements such as beliefs, knowledge, confidence, perceived ability, and attitude. Previous research has concluded that contextual factors exist on a macroscale and could include influences from within a person’s close living environment, as well as broader factors such as urban structure or the political situation of the state [38,39,40]. The evidence above highlights the intricate and significant interaction between different influencers on PA and health behaviour, demonstrating how sociodemographic factors, such as gender and socioeconomic status, alongside environmental factors are all intertwined, collectively guiding human behaviour and decision-making. Subsequently, to understand the causes of unhealthy behaviours, such as inactivity or sedentary behaviour, a social ecological approach must be adopted, acknowledging the broader effects of key influencers, as opposed to solely focusing on independent factors such as access to parks or the provision of physical education (PE) lessons. Previous research has reported that the one-size fits all approach in promoting PA/PE is ineffective because of the different social, environmental, and cultural barriers that young people from a range of demographics experience. For instance, the social influence is interlinked with positively or negatively affecting people’s participation in PA. People are more likely to engage in PA or sports in countries where PA is considered to be important for health and wellbeing compared to locations where PA is not a social norm [41]. Furthermore, the social influences towards PA and environmental differences were also postulated as explanations for British participants being more active compared to counterparts in Saudi Arabia [7,42]. Moreover, the environmental influence characteristics also play a role in youth PA and sport participation. For instance, young people who live within close distance to a park were more likely to be classified as healthy weight compared to children without nearby parks [43]. Previous research also reported that neighbourhood residents who have access to supermarkets and limited access to convenience stores tend to have healthier diets and lower levels of obesity [44]. Subsequently, adopting a social ecological approach helps to explore various factors influencing people’s behaviour. There is sparse research exploring the influence of PA and health behaviour in Saudi Arabian youth, and this further underlines the need for research underpinned by the SEM to develop a greater understanding of how PA and healthy lifestyles can be promoted within a region that is currently rife with inactivity, cardiovascular disease, and obesity. Therefore, the aim of this research was to explore the factors influencing the PA engagement of youth from different geographical locations in the KSA from the SEM perspective. Thus, the SEM is an appropriate theoretical approach for this study, as it considers both individual and environmental factors.

## 2. Methods

### 2.1. Participant Recruitment and Procedures

Following an institutional ethical approval, information letters with details about the study were sent to schools within the Al-Ahsa governorate, located in the Eastern province of the KSA. Stratified opportunity samples of schools were selected according to their geographical locations (urban, rural farm, and rural desert). Within each of the three stratified categories, one school of males and one school of females were invited to participate. After recruiting eight schools, invitations were sent to all students in selected schools and information was relayed to parents and students with details of the study. The consent was gained from parents for their children to participate in this study. Sixteen focus groups (male focus groups = 8 and female focus groups = 8) were conducted with a total of 120 secondary school individuals (male = 63 and female = 57) aged between 15 and 19 years. Two focus groups were conducted per school, with a group size between 8 and 12 participants. The number of participants in each group was within the recommended parameters based on previous studies [45,46].

### 2.2. Focus Groups: Practical Considerations and Procedures

Participants were advised prior interviews that their conversations would be recorded, and all the responses given were implicitly confidential. In addition, respondents were able to leave the interview at any time without question. The female focus group sessions were conducted by a research assistant who had attended a training course with regard to focus group facilitation at the King Faisal University and hence had the necessary skills to conduct the sessions. Considering the chronological age of the participants, great care had to be taken when planning the duration of the focus group. This is because young people have limited attention spans, and communication skills over a long period may be impeded [47,48]. Therefore, sessions were intended to be 45 min in length but could be shortened or increased subject to the progression of the group and in any state of the respondent. For instance, the session could be ended earlier if the respondents showed signs of fatigue. The group sessions were recorded through digital audio device (Sony ICD-UX200), with the researcher transcribing, reviewing, and analysing materials recorded. The confidentiality was obtained, parents and participants were assured that during the analysis and subsequent presentation of the data, their responses would be anonymous by replacing participants’ names with ID numbers in the transcripts. The sessions were effectively executed by a semi-structured plan and questions posed to the participants are outlined in Table 1.

### 2.3. Pilot Focus Group

The questions were piloted with one group of male youth and one group of females. Six male and four female participants were invited to attend these pilot groups. The female group was observed by the lead researcher who provided feedback to the staff-trained female at the end of the group. There were some areas to highlight during the piloted study. For instance, there was an occasion when the staff trained did not acknowledge or respond to a direct question from a participant. Secondly, there was an example of a positive use of probing questions. These were highlighted and communicated with the staff trained and were subsequently overcome during the final focus groups. Throughout the pilot sessions, it was noted that many respondents felt that the best way to become involved in the discussion was to raise their hand as the researchers came to the end of a discussion. It was agreed that this did not allow the discussion to flow as intended, and the sessions became more similar to a school lesson than an open discussion. To counter this, group guidelines were established with the aim of encouraging participants, promoting respect for other people’s views, and emphasising that this was not a test and that there were no ‘wrong’ responses.

### 2.4. Data Analysis

The focus groups were analysed to identify the key themes through the process of transcribing by applying a thematic analysis (TA). The utilisation of TA in qualitative research is common because it offers a systematic and transparent structure for data analysis [49]. Previous research used and supported the application of TA for exploring surrounding issues [46,50,51,52]. Therefore, for the purpose of this research, TA was deemed appropriate. Following the interviews, the framework of the focus groups were transcribed verbatim, checked, and coded. Subsequently, regarding the trustworthiness of the results, the common model, which consisted of five conditions, i.e., credibility, dependability, conformability, transferability, and authenticity, were combined to construct the trustworthiness and applied to our research. The data were member-checked from the beginning (raw data) until the completion of analysis for data saturation.

## 3. Results

This sample covers three different geographical locations in the Al-Ahsa governorate in the KSA. The geographical locations were urban, rural farm, and rural desert. The characteristics of 16 focus groups members are presented in Table 2. It was considered beneficial to allow and encourage interactions between participants. This is because it has been shown that one of the main advantages of focus groups is that it allows for the participants’ natural exuberance, thereby facilitating the willingness to interact and express opinions. Therefore, this was a contributing factor in selecting this method. There was a range of influential factors affecting youth PA from across all three geographical locations and suggested facilitators, including the emergence of themes and raw data that contributed to the focus groups. For instance, the awareness of PA provided by the participants incorporated three subthemes, including the meanings of PA, types of PA, and perceived benefits (psychological, sociological, and health-related). Additionally, the different types of activity between genders were noted. For example, football, jogging, and gym were the most cited type of PA among males, whilst walking was more favoured among female participants. Moreover, the environment was the prevailing issue mentioned by all groups that could influence their PA participation. This included cultural norms as participants stated that this plays an important role in discouraging youth and, in particular, females from participating in PA in open areas.

Although this theme was reported frequently in all groups from different geographical locations, it was most evident among the rural desert youth. Many participants highlighted the lack of importance attached to and the negative attitude towards sports and PA from a societal point of view. For instance, the majority of females, particularly those from rural desert areas, reported that social and cultural beliefs do not allow them to partake in sports and PA in open areas and it is against cultural norms. Females also reported that the community (society) looks at women who participate in outdoor sports in a disagreeable light, even if they wear the abaya and hijab. Hence, from society’s perspective, females must only perform sports and PA in private areas (female-only) or in their homes. There was an agreement among participants’ responses that they were not satisfied with the sports and recreational facilities provided in their neighbourhoods. The absence or the lack of sports and/or recreational facilities, their poor quality (inferior) and being too far from such facilities were cited as barriers to PA and sports participation among all groups. When asked ‘what can help promote PA levels’ the participants from all groups identified several strategies and suggestions that could improve PA levels. These suggestions were related to neighbourhood characteristics, cost of activities, physical education lessons, and the awareness of the importance of PA. In total, four themes were identified, i.e., intrapersonal factors, interpersonal factors, environment, community, and organization, and the suggested (strategies) for policymakers to enhance PA were proposed alongside the subthemes and raw data for each theme, as outlined in Table 2.

## 4. Discussion

In the current study, although most of the Al-Ahsa youth reported not being regularly active, there were differences between gender and geographical location in PA level and types of activities. For instance, females were reported to be less active than males, and participants from rural desert locations were also reported to be less active compared to other geographical areas. The current findings align with previous studies of a similar nature [8,42,53]. Furthermore, results regarding social ecological determents to Al-Ahsa youth participating in PA were interesting. For instance, football as PA was favoured by male youths, whereas walking was preferred by females. The present findings support previous research suggesting that males prefer to partake in weightlifting, team, and competitive sports, and the females’ PA preference is identified to be walking and yoga [54,55]. In addition, participants in this study were aware of the benefits of engaging in sport or exercise, yet most of them did not partake in regular PA.

The common intrapersonal influences on youth PA identified in this study included: “negative attitude towards sports and physical activity”, “laziness and lack of motivation”, “lack of time” and “cost of activity”. The intrapersonal factors were identified as “knowledge, attitude and motivation” as outlined in (Table 2). However, the intrapersonal and intrapersonal factors influencing young people’s PA have been investigated [56,57,58,59,60]. Nevertheless, the present findings contribute to the existing literature regarding the barriers and its correlation between knowledge and PA participation of KSA youth. The current findings align with previous research suggesting that a lack of motivation, laziness, attitude, and preferences of young people across different countries impact their PA engagement [61,62,63]. Despite the intrapersonal and interpersonal factors, the youth in KSA have been found to continue to make the effort to partake in some sports or PA. Previous research reported that the youth are more inclined to participate in sports or exercise when it is tailored to their preferences, and this enables them to make choices and an ongoing commitment for PA or sport participation to be part of their lifestyle routine [61,64]. Future research is recommended to focus on the under-researched population to increase their knowledge and positive attitude towards sports and PA participation, and this could be performed through the PE curriculum in schools.

In the current study, participants reported that they are in school during the day and busy completing their homework and preparing for exams in the evening. Therefore, they do not have enough time to participate in sports or PA. For instance, as suggested by a participant, ‘I was playing sports in the past, but now I do not have enough time. I am busy in my study and exams, so I do no not have time to go to the gym. There is no time at all, so I stopped sports.’ This finding aligns with previous research suggesting that lack of time is one of the common barriers to PA participation [46,65]. Although many participants in this study felt that they do not have enough time to partake in PA or sports, most of them—especially females—were reported to spend a high amount of their time watching television and using the internet. In contrast, most of the male participants reported to be spending more time with friends in a social gathering. It can be argued that future studies could explore beyond the barriers to PA or sport participation. For instance, interventions and strategies can be focused on reducing screen time and, as an alternative, offer opportunities for young people to participate in PA or sports by tailoring it to their preferences. Previous research has reported that watching television, playing console games, and spending prolonged hours on other sedentary behaviour are major contributing factors to the inactive epidemic in the KSA and worldwide [2,66,67,68,69].

The cost of using sports centres, including gym memberships, were commonly reported as a barrier contributing to the inactive lifestyle of the under-researched population. This was particularly a barrier for female participants. The high cost of organised structured sports and exercise has been associated with decreased PA levels, especially amongst females and for individuals from a lower socioeconomic status [46,70]. Moreover, it was found that people who reported cost as a barrier to PA or sport participation also suggested lack of time and low awareness about the benefits of partaking in regular PA and were less likely to join the organised activity classes. Young people are more likely to engage in activities with easy access and low cost with an availability of facilities at their discreet [71]. Therefore, future interventions focused on promoting PA through the provision of free sports and PA facilities could be effective to the Al-Ahsa youth. 

At the second level of the social ecological model, the current study identified interpersonal factors, such as cultural norms and beliefs/social support. These factors referred to the relationships between youths and those who influenced their behaviours (e.g., parents, peers, significant others). Previous studies indicated that social norms are important predictors of health behaviours such as PA and healthy lifestyle [41,72]. In the current study, social norms and cultural beliefs were reported frequently by all focus groups as a barrier to participate in outdoor sports and PA. This was obvious for both males and females from all geographical locations. The negative attitude towards sports, PA, and gender inequality within Saudi society were major influences, specifically on female PA. For example, many people within the Saudi community consider females who participate in basic exercise, such walking outdoors, in an unfavourable light, even if one wears the abaya and the hijab. The current findings support previous research by [41], concluding that Afghan females view the lack of single-sex facilities and the requirement to be fully covered as important barriers to their PA participation. Nevertheless, socially imposed obstacles can negatively influence PA participation for both genders. For instance, in this study, the social norms were the key influence among females, and this also impacted males, particularly those from the rural desert areas. For example, males from the rural desert reported that wearing sportswear is unacceptable in a rural desert (Bedouin) society. The current findings align with previous research, suggesting that wearing exercise clothes are seen as a Western culture and is regarded as a barrier to PA participation in Afghanistan but not, for example, in the United Kingdom [41].

In the present study, the paternal role, practices, and support were also reported as a major factor of influencing young people’s sports and PA participation, particularly for females. Most of the participants described their parents as physically inactive. For instance, “*my family always disappoints me, with their view of sports as a waste of time, they say to me, why do you do sports instead of study*”. Previous studies revealed that parents can influence their children’s lifestyle, including health and sports participation [73,74,75]. The importance of parental support is undisputable, but in the current study, participants reported not receiving any type of parental support or encouragement to partake in sports or PA. However, it should be noted that the concerns of parents reported by participants were different for males and females. For instance, in males, concerns were study demands and fear of injury, whereas for females, their parents’ concerns were of the unsafe neighbourhoods and family honour. Previous research also suggested that parents living in areas of high physical disorder and lower (e.g., graffiti, rubbish, and low greenery) perceived safety as the main reason to less likely encourage their children to participate in outdoor activities. This is an example of the direct influence of the characteristics that a built environment can influence the safety perceptions of adults and subsequently impact their interaction with children to either encourage or deter PA within that environment. Some of the other potential reasons that parents in the KSA are not supportive of sports or PA participation could also be due to the lack of time, awareness of PA benefits, and culture. Previous research suggested that lack of time, awareness, and cultural expectations and acceptance of participating in sports or PA vary across countries and communities [46]. For instance, countries where sports or PA is considered as vital for health and wellbeing are prone to recognise fewer barriers to PA; and therefore, engagement in regular PA is higher compared to countries where PA is not a social norm [41]. The cultural attitude towards PA and environmental differences were also postulated, and findings indicated that British participants were more active compared to counterparts in Saudi Arabia [7]. Instead of playing sports or engaging in PA, young people in the current study reported to be spending their free time with friends for social activities such as playing computer games in a sedentary behaviour, which could lead to obesity and diabetes.

The current findings also highlighted environmental and communal factors, including neighbourhood characteristics such as recreational and sport facilities, sidewalks, neighbourhood, school environment, home environment, and weather. Although a lack of sport and recreational facilities were reported frequently by participants across all focus groups, this was particularly obvious among the youth from the rural desert. It was found that the number of nearby recreation facilities and parks were associated positively with girls’ PA, whereas retail-store ratio correlated positively with boys’ PA levels. Furthermore, proximity to facilities were also highlighted by some participants, as the need for transportation to reach the nearest sidewalks and sports facilities encumbered their willingness to perform any PA. For instance, one of the nearest sport or sidewalk facility was within five to ten miles and one would need a car to cover the distance and participate in sports or PA. Distance to facilities can influence youth PA, particularly for females. For instance, Potwarka and Kaczynski [43] found that children who lived within one kilometre of a park or playground were almost five-fold more likely to be classified as healthy weight compared to those without playgrounds in neighbouring parks. Furthermore, visitors who were observed in park environments, which contained playgrounds, sport facilities, and paths were significantly more active than visitors in settings without these features [76]. Improving the neighbourhoods’ characteristics by providing a variety of sports and recreational facilities is needed to enhance the PA levels among the youth in Saudi Arabia.

Participants also identified school environment and policies as barriers to PA participation. In the current study, schools were identified as the main organisation for promoting PA and healthy behaviour. Young people spend a considerable amount of time at school; therefore, schools could offer excellent opportunities for young people to engage in PA, and this could be performed through PE classes, during recess periods, and travel time to school. Most of the participants were not satisfied with the sports facilities in their schools; this included the absence and lack of sports fields, poor quality of sports equipment, and an absence of an indoor playground. The school sports facilities can influence the PA levels of young people. Previous research has concluded that school playground markings and availability of sports equipment is associated with children and youth PA [77,78]. Therefore, providing access to facilities and redesigning the playground may help increase recess PA among KSA young people.

The geographical climate, such as hot weather, was reported by participants to be a barrier to outdoor PA. While previous research highlighted that young people can be influenced by built environment, the weather has been overlooked in the literature [79]. The current findings are in contrast with previous studies, concluding that young people’s outdoor PA levels decreased during the winter period, especially in countries such as the USA and the UK [80]. On the other hand, the summer season, with extremely hot weather during most of the year in the KSA can be a major influence on youth PA reduction. Participants also reported that they cannot play outside during most of the daytime and only play for a short time post-sunset. This limits the time for individuals to perform outdoor physical activities and sports. The present findings reinforce the need to improve access to facilities during summer for the under-researched population to participate in regular PA and sports, otherwise, their health and wellbeing could be negatively affected.

## 5. Strengths and Limitations

This study found its strength in evaluating the factors influencing PA engagement among youth from different geographical locations in the KSA from a SEM and qualitative approach. To the best of the authors’ knowledge, this is one of the fewest (if not the first) study focused on exploring factors and their impact on the youth in the KSA. This study contributes to the rare research and further highlight the need for future research and policy interventions to address the concerns of this population. Despite the strength, this study is not without limitations. Limitations included participants from one region, Al-Ahsa, and targeted youths only. Collecting data from other regions across the KSA and varying age groups may have yielded different results. The additional limitation of this study was that the PA data were not collected and only discussed during the focus groups. Collecting PA data through objective tools, providing statistical analyses of PA levels, and conducting a comparison between gender and locations would have yielded different results. Therefore, future research is warranted to gather PA data via objective methods such as ActiGraph and provide statistical analyses and differences (if any) between gender and locations of the under-researched population.

## 6. Conclusions

This is one of the first studies to use a SEM to better understand the multiple potentials that influence the KSA young people’s PA participation. This study found that females were less likely to engage in PA or play sports compared to males. The current findings also showed that the social norms and cultural beliefs had a greater negative influence on females’ PA than males. Findings also highlighted that the influence of intrapersonal, interpersonal, communal, and environmental factors positively and/or poorly impact youth PA, especially among females. This study further contributes to the knowledge and understanding of the complex interplay of the multifactors that influence KSA youths’ PA behaviour qualitatively. By exploring the different factors influencing young people’s PA behaviour through the SEM, the current understanding of the multidimensional effects that affect the under-researched population in the KSA from different geographical locations has expanded. This qualitative approach has provided the participants a voice and offer valuable evidence to policymakers, public health departments, and local authorities with regard to PA intervention based on the environment and the community.

## Figures and Tables

**Table 1 ijerph-20-05785-t001:** Questions posed to participants during focus group interviews.

Question 1.	Explored the participants’ definitions of ‘PA’, including their attitudes towards PA and awareness of PA benefits.
Question 2.	Assessed the participants’ perceptions of their home environment; this included sedentary behaviour and domestic sport equipment.
Question 3.	Assessed the participants’ perceptions of their school environment, including PE lessons and sport facilities in the school.
Question 4.	Assessed the participants’ perceptions of social influences; this included family, peers, and societal influences.
Question 5.	Assessed the participants’ perceptions of neighbourhood characteristics, such as sports and recreational facilities, access to walkways, availability of parks, and access to cycling paths.
Question 6.	Identified the barriers and motivations reported by the respondents when participating in PA, and the suggestions to promote PA from their points of view.

**Table 2 ijerph-20-05785-t002:** A summary of themes and examples of raw data extracts from the focus groups from the three geographical locations.

Themes	Selected Quotes from Participants
Intrapersonal Factors	The individual factors discussed were varied as outlined
Awareness/definition of PA	M8: ‘moving the body or any part of the body could be PA’ (Urban school).
	F37: ‘anything that activates blood circulation, I consider it as PA, even walking, or things that I do such as housework, for example, cleaning and washing or any things that require movement and stimulate blood circulation’ (Rural farm school).
Participating in PA	M14: ‘it is always available and affordable, does not need special places and equipment; you can play football anywhere’ (Urban school).
	F16: ‘I do not do sports, but I only do it when I go for shopping. Hahah (group laughing). It is true, we all do walk exercise when we do shopping’ (Urban school).
Perceived health benefits	M52: ‘building up muscle reduces joint diseases and osteoporosis; a person who practices sport and PA at the age of 50 years or 60 years looks like someone of the age of 20 or 30 years’ (Rural desert school).
	F38: ‘sports are the most important thing, it protects us from disease and reduces weight, and even as we get older, playing sports postpones aging symptoms’ (Rural farm school).
Negative attitudes towards PA	M36: ‘I do not like sports…I do not like to do sports and wear sports uniform since I was a child… I prefer meeting friends and talking with them instead of playing sports’ (Rural farm school).
	F56: ‘if I start sports and then suddenly stopped doing sports, I will gain more weight than before I started’ (Rural desert school).
Lack of time	F25: ‘I was playing sports in the past, but now I do not have enough time. I am busy in my study and exams, so I do no not have time to go to the gym. There is no time at all, so I stopped sports’ (Urban school).
	F40: ‘I used to do exercise almost on a daily basis; now I cannot have time for exercise, I do help my mother with housework and then I study, and I do my homework, then sleep’ (Rural farm school).
Laziness	M25: ‘laziness and sluggish, especially after having meals. This time person looking for the TV and to sit’ (Urban school).
Freedom	F13: ‘boys can go out with their friends to exercise, but for girls, even if her family allows her to go out, her friends’ families do not agree for girls to go out for sports. So, the girl loses the companionship advantage in sports and encouragement’ (Urban school).
	F9: ‘the community in which I live; they have only sports for boys, girls cannot go out and do sports, they look at her as if she is doing a big misdemeanour’ (Urban school).
Cost	F22: ‘society does not encourage doing sports. There are no sports clubs especially for girls, and if some were available, they will be too expensive. Many girls who want to go to the sports club cannot go due to the high cost of membership’ (Urban school).
	M22: ‘football field fees vary from field to another, but in general the fee is around 250 Saudi riyal… for middle school students cannot afford the fee so they play in the street, but for high school students it is a reasonable price for us’ (Urban school).
Interpersonal Factors	Social environment was the prevailing issue mentioned by groups that could influence their PA
Cultural and social norms	M62: ‘we live in the desert area, we do not live in the city such Al-Ahsa or Al-Kharj, we are a well-known family, and we are famous in camels; we cannot do sports’ (Rural desert school).
	F9: ‘even if I wanted to go and do sports, they all said you are thin, you do not need sports, also all females who go to the gym, they are adult females. There are no girls my age’ (Urban school).
	F39: ’this is depending on the family beliefs more than village beliefs. Some families agree for girls to go out to walk and some families do not, some families are very strict in their beliefs and culture norms, and some are normal’ (Rural farm school).
	M62: ‘I do not wear sports uniform. From society’s view, you have to wear Thoub and Ghotrah (traditional clothes). If you wear sports uniform you will be a ‘bad boy’, even if you wear normal shirt and pants to the mall people will look at you in a bad way’ (Rural desert school).
	M54: ‘we are Bedouin, (moderator: so what); it is a shame on us if we wear sports kit’ (Rural desert school).
Social support	M33: ‘my father does not do sports, because he has something better than exercise, for example, he reads books, He believes that reading is more important than sports’ (Urban school).
	F18: ‘my family always disappoints me, with their view of sports as a waste of time, they say to me, why do you do sports instead of study’ (Urban school).
	M40: ‘sometimes I play football and meet friends for fun’ (Rural farm school).
Environment, Community, and Organisation	Participants’ perceptions indicated several environmental factors that play an important role in influencing their PA
Neighbourhood characteristics	M25: ‘there is only one football field playground, and it is shared by many neighbourhoods’ (Urban school).
	F18: ‘I want to know why these places are not for girls? Why are all these places just for boys and not girls? Until when we are like this situation? Listen, government will provide places and sports facilities, but the problem is in society, the society is sick’ (Urban school).
School environment/facilities	M52: ‘Physical education is 45 min, first 5 min for changing and dressing, after you just start play, we stopped 5 or 10 min before for changing and dressing again and smell badly of sweat’ (Rural desert school).
	M36: ‘I do not like playing football, physical education is just football’ (Rural farm school).
	M16: ‘if there is changing room available and indoor playground, we will wear sports uniform and play. However, now, where we can change? In the class or the playground?’ (Urban school).
Home environment	M10: ‘I use the TV for PlayStation more than to watch TV channels’ (Urban school).
	F46: ‘when I am in the living room I am on the TV, I mean all my time in the house will be on the TV’ (Rural farm school).
	M26: ‘for social networking, Facebook and Twitter, and sometimes I use the internet to search information for homework’ (Urban school).
	F14: ‘female websites, fashion, cooking and others’ (Urban school).
Weather	M52: ‘the weather in the desert extremely hot, no one can go out during the day, the sun will burn it ha ha ha (laughing)’ (Rural desert school).
	M10: ‘depending on the weather, if in the winter we play during daytime, but in summer I prefer at night’ (Urban school).
	M45: ‘at the end of the daytime the sun’s heat becomes mild, so we can play’ (Rural farm school).
Safety	M52: ‘Sometimes if you walk at night, it is possible someone hits you with his car when you walk, he could be drunk or on drugs, and you can be the victim’ (Rural desert school).
	F50: ‘if you are with your father or brother, it is fine, but alone someone could harass you’ (Rural desert school).
Suggestions (Strategies) for Policymakers to Enhance PA	The participants from all groups identified several strategies and suggestions to improve physical activity for people, as outlined below:
Neighbourhood characteristics	M19: ‘the city council should establish sports fields for each neighbourhood and should run sports neighbourhoods’ leagues’ (Urban school).
	F2: ‘increase the number of women’s sports clubs. These clubs should be integrated with different sports activities in all neighbourhoods. We have one small gym inside the beauty shop’ (Urban school).
	F39: ‘sidewalks, and I hope to increase spaces and places for the exercise’ (Rural farm school).
Cost of activities	F25: ‘If there is a sports centre for each neighbourhood, and family membership for families. And fees for membership but not expensive, these fees go to coaches and maintenance, and beside this centre a small park. A person who has membership can use the centre and do sports’ (Urban school).
Awareness of the importance of physical activities	F55: ‘Increase the awareness of sports for mothers and fathers and society which results in them encouraging their children to do sports’ (Rural desert school).
	F18: ‘Before focusing on a girl, we should focus on her house and family. And increase their awareness of sports. This can be from school or the TV or flyers and brochures, then we can focus on girls’ (Urban school).

## Data Availability

Not applicable.

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
