# Peer review of "Factors Influencing Saudi Youth Physical Activity Participation: A Qualitative Study Based on the Social Ecological Model"

_ijerph, 2023, doi:10.3390/ijerph20105785_

Round 1
Reviewer 1 Report (Previous Reviewer 1)
- The first point is that the references at the end of the article are not based on journal format.
- In the introduction, the social ecological models is not properly explained. Please explain the authors of this model more. Has this model been investigated in other countries? Please clarify.
- Results are much improved. Thank you.
- Discussion and conclusion are good.
Best,
Minor editing is needed.
Author Response
Dear Reviewer
International Journal of Environmental Research and Public Health
Thank you for reviewing our manuscript which is prepared for the above journal.
We are pleased to have completed necessary amendments and have uploaded this new version online as requested. We also attach responses to your comments in the table below.
Please let us know if we can provide you with any further information. Thank you for your time.
Yours sincerely,
Dr Ayazullah Safi

This manuscript is a resubmission of an earlier submission. The following is a list of the peer review reports and author responses from that submission.
Round 1
Reviewer 1 Report
- The first point is that the references in the text and at the end of the article are not in the magazine format. Please follow the magazine format throughout the article.
- In the introduction, the second paragraph is not very relevant to study. What kind of means do children and teenagers travel with, or what happens during the time from childhood to adulthood in the transportation of these people, what is the relationship with social factors? These factors are related to the environment, not social factors.
- Then, the social ecological models is not properly explained. Please explain the authors of this model more. Has this model been investigated in other countries? Please clarify.
- Overall, the introduction needs revision. In the introduction, we should focus more on the social ecological model and social factors affecting youth participation in physical activity.
- The research method and presentation of results in the current research also have technical problems. Firstly, qualitative research method has been used in the current study. Qualitative research method generally ends with providing a model of influencing factors. Where is this model in the research? The authors can present the common components that have been extracted from the interviews in the form of a model of influential social factors. However, no model is presented in the report of the results of this study. Therefore, it is not clear what the results of the interviews are.
- Regarding the use of the questionnaire, the question arises, what is the role of this questionnaire and its results in this study? Knowing how many percent of the subjects had low or high physical activity, what effect does it have on identifying social factors affecting their participation in physical activity? How are the results of the questionnaire related to the results of effective social factors? Why are they measured and reported at all? This issue is not clear at all.
- Finally, qualitative research can be accompanied by a quantitative study. For this purpose, the components identified from the interviews are placed in the form of a questionnaire and this questionnaire is distributed among the participants and the results are presented as a quantitative model of social factors affecting physical activity. This work was not done in the current study and this can be considered as a weakness of this study.
Author Response
Dear reviewer 1 many thanks for reviewing our paper. Please see the responses attached.

Reviewer 2 Report
This is a topical work, based on the situation of the deterioration of the practice of physical activity among young people.
The work is novel in several aspects: first, it offers a vision of what is happening in a specific country and group (Saudi youth) and for its theoretical and methodological proposal.
Issues for improvement:
The title: with a qualitative methodology it is difficult to generalize, it should be added that they are case studies.
In the abstract, because it uses a qualitative methodology, more emphasis should be placed on this aspect in the results, rather than on the quantitative data.
Change backgraund to introduction.
In this section, a clear differentiation should be made between sports practice within the framework of physical education in schools and outside, i.e., in leisure time.
This differentiation should be clear in the ecological model and in the results.
Method
It develops the application of a qualitative methodology in a comprehensive manner.
But also, in dissonance with the title, it recognizes the application of the Physical Activity Questionnaire (quantitative instrument). And its application and analysis is not done in depth.
How it was administered, with what statistical program it was treated, what statistical analysis they perform, etc.
Results:
The results are mainly reduced to a table, without a commentary, previous from the authorship, also with this presentation, it is not possible to appreciate the possible divergences in the discussion of the focal group.
Quantitative data from the questionnaire are not developed.
Discussion:
Adequate, but does not address the differences, but rather the similarities within each factor. Nor does it take into account the context: free time or in the regulated framework of the educational system.
In summary, it is an interesting, novel and publishable work if it undertakes part of the suggestions or nuances suggested.
Author Response
Dear Reviewer 2, thank you for reviewing our paper. Please find attached the responses.

Round 2
Reviewer 1 Report
Dear authors
Thanks for correcting the paper. In my opinion, the paper has technical problems in terms of research method and presenting of results. Statistically, the relationship between physical activity data and interview data is not clear. Overall, the results are mostly subjective rather than based on statistical analysis.
Thank you again for your revision.
Author Response
Dear Reviewer 1:
|
Thank you for this observation. Due to the nature and the aim of this study, we have collected the physical activity data through one of the most valid and reliable tools that exist (IPAQ) and qualitative data was gathered via focus groups. The descriptive statistics for physical activity as mean and SD are outlined and the themes, sub-themes and raw data had also been explored and discussed in detail. Due to the nature of this study, we do not think there was a need for statistical tests. Thank you again for your review and much appreciated. |

Reviewer 2 Report
The author of the text has made the suggested changesAuthor Response
Thank you for taking time out and reviewing our paper again. Much appreciated